# Conditional Adversarial Domain Adaptation

**Mingsheng Long[†], Zhangjie Cao[†], Jianmin Wang[†], and Michael I. Jordan[♯]**
[†]School of Software, Tsinghua University, China
[†]KLiss, MOE; BNRist; Research Center for Big Data, Tsinghua University, China
[♯]University of California, Berkeley, Berkeley, USA
{mingsheng, jimwang}@tsinghua.edu.cn  caozhangjie14@gmail.com
jordan@berkeley.edu

## Abstract

Adversarial learning has been embedded into deep networks to learn disentangled and transferable representations for domain adaptation. Existing adversarial domain adaptation methods may not effectively align different domains of multimodal distributions native in classification problems. In this paper, we present conditional adversarial domain adaptation, a principled framework that conditions the adversarial adaptation models on discriminative information conveyed in the classifier predictions. Conditional domain adversarial networks (CDANs) are designed with two novel conditioning strategies: multilinear conditioning that captures the cross-covariance between feature representations and classifier predictions to improve the discriminability, and entropy conditioning that controls the uncertainty of classifier predictions to guarantee the transferability. With theoretical guarantees and a few lines of codes, the approach has exceeded state-of-the-art results on five datasets.

## 1 Introduction

Deep networks have significantly improved the state-of-the-arts for diverse machine learning problems and applications. When trained on large-scale datasets, deep networks learn representations which are generically useful across a variety of tasks [36, 11, 54]. However, deep networks can be weak at generalizing learned knowledge to new datasets or environments. Even a subtle change from the training domain can cause deep networks to make spurious predictions on the target domain [54, 36]. While in many real applications, there is the need to transfer a deep network from a source domain where sufficient training data is available to a target domain where only unlabeled data is available, such a transfer learning paradigm is hindered by the shift in data distributions across domains [39].

Learning a model that reduces the dataset shift between training and testing distributions is known as domain adaptation [38]. Previous domain adaptation methods in the shallow regime either bridge the source and target by learning invariant feature representations or estimating instance importances using labeled source data and unlabeled target data [24, 37, 15]. Recent advances of deep domain adaptation methods leverage deep networks to learn transferable representations by embedding adaptation modules in deep architectures, simultaneously disentangling the explanatory factors of variations behind data and matching feature distributions across domains [12, 13, 29, 52, 31, 30, 51].

Adversarial domain adaptation [12, 52, 51] integrates adversarial learning and domain adaptation in a two-player game similarly to Generative Adversarial Networks (GANs) [17]. A domain discriminator is learned by minimizing the classification error of distinguishing the source from the target domains, while a deep classification model learns transferable representations that are indistinguishable by the domain discriminator. On par with these feature-level approaches, generative pixel-level adaptation models perform distribution alignment in raw pixel space, by translating source data to the style of a target domain using Image to Image translation techniques [56, 28, 22, 43]. Another line of works align distributions of features and classes separately using different domain discriminators [23, 8, 50].

Despite their general efficacy for various tasks ranging from classification [12, 51, 28] to segmentation [43, 50, 22], these adversarial domain adaptation methods may still be constrained by two bottlenecks. First, when data distributions embody complex multimodal structures, adversarial adaptation methods may fail to capture such multimodal structures for a discriminative alignment of distributions without mode mismatch. Such a risk comes from the equilibrium challenge of adversarial learning in that even if the discriminator is fully confused, we have no guarantee that two distributions are sufficiently similar [3]. Note that this risk cannot be tackled by aligning distributions of features and classes via separate domain discriminators as [23, 8, 50], since the multimodal structures can only be captured sufficiently by the cross-covariance dependency between the features and classes [47, 44]. Second, it is risky to condition the domain discriminator on the discriminative information when it is uncertain.

In this paper, we tackle the two aforementioned challenges by formalizing a conditional adversarial domain adaptation framework. Recent advances in the Conditional Generative Adversarial Networks (CGANs) [34, 35] disclose that the distributions of real and generated images can be made similar by conditioning the generator and discriminator on discriminative information. Motivated by the conditioning insight, this paper presents Conditional Domain Adversarial Networks (CDANs) to exploit discriminative information conveyed in the classifier predictions to assist adversarial adaptation. The key to the CDAN models is a novel conditional domain discriminator conditioned on the cross-covariance of domain-specific feature representations and classifier predictions. We further condition the domain discriminator on the uncertainty of classifier predictions, prioritizing the discriminator on easy-to-transfer examples. The overall system can be solved in linear-time through back-propagation. Based on the domain adaptation theory [4], we give a theoretical guarantee on the generalization error bound. Experiments show that our models exceed state-of-the-art results on five benchmark datasets.

## 2   Related Work

Domain adaptation [38, 39] generalizes a learner across different domains of different distributions, by either matching the marginal distributions [49, 37, 15] or the conditional distributions [55, 10]. It finds wide applications in computer vision [42, 18, 16, 21] and natural language processing [9, 14]. Besides the aforementioned shallow architectures, recent studies reveal that deep networks learn more transferable representations that disentangle the explanatory factors of variations behind data [6] and manifest invariant factors underlying different populations [14, 36]. As deep representations can only reduce, but not remove, the cross-domain distribution discrepancy [54], recent research on deep domain adaptation further embeds adaptation modules in deep networks using two main technologies for distribution matching: moment matching [29, 31, 30] and adversarial training [12, 52, 13, 51].

Pioneered by the Generative Adversarial Networks (GANs) [17], the adversarial learning has been successfully explored for generative modeling. GANs constitute two networks in a two-player game: a generator that captures data distribution and a discriminator that distinguishes between generated samples and real data. The networks are trained in a minimax paradigm such that the generator is learned to fool the discriminator while the discriminator struggles to be not fooled. Several difficulties of GANs have been addressed, e.g. improved training [2, 1] and mode collapse [34, 7, 35], but others still remain, e.g. failure in matching two distributions [3]. Towards adversarial learning for domain adaptation, unconditional ones have been leveraged while conditional ones remain under explored.

Sharing some spirit of the conditional GANs [3], another line of works match the features and classes using separate domain discriminators. Hoffman *et al.* [23] performs global domain alignment by learning features to deceive the domain discriminator, and category specific adaptation by minimizing a constrained multiple instance loss. In particular, the adversarial module for feature representation is not conditioned on the class-adaptation module with class information. Chen *et al.* [8] performs class-wise alignment over the classifier layer; i.e., multiple domain discriminators take as inputs only the softmax probabilities of source classifier, rather than conditioned on the class information. Tsai *et al.* [50] imposes two independent domain discriminators on the feature and class layers. These methods do not explore the dependency between the features and classes in a unified conditional domain discriminator, which is important to capture the multimodal structures underlying data distributions.

This paper extends the conditional adversarial mechanism to enable discriminative and transferable domain adaptation, by defining the domain discriminator on the features while conditioning it on the class information. Two novel conditioning strategies are designed to capture the cross-covariance dependency between the feature representations and class predictions while controlling the uncertainty of classifier predictions. This is different from aligning the features and classes separately [23, 8, 50].

# 3 Conditional Adversarial Domain Adaptation

In unsupervised domain adaptation, we are given a source domain $\mathcal{D}_s = \{(\mathbf{x}_i^s, \mathbf{y}_i^s)\}_{i=1}^{n_s}$ of $n_s$ labeled examples and a target domain $\mathcal{D}_t = \{\mathbf{x}_j^t\}_{j=1}^{n_t}$ of $n_t$ unlabeled examples. The source domain and target domain are sampled from joint distributions $P(\mathbf{x}^s, \mathbf{y}^s)$ and $Q(\mathbf{x}^t, \mathbf{y}^t)$ respectively, and the i.i.d. assumption is violated as $P \neq Q$. The goal of this paper is to design a deep network $G : \mathbf{x} \mapsto \mathbf{y}$ which formally reduces the shifts in the data distributions across domains, such that the target risk $\epsilon_t(G) = \mathbb{E}_{(\mathbf{x}^t, \mathbf{y}^t) \sim Q} [G(\mathbf{x}^t) \neq \mathbf{y}^t]$ can be bounded by the source risk $\epsilon_s(G) = \mathbb{E}_{(\mathbf{x}^s, \mathbf{y}^s) \sim P} [G(\mathbf{x}^s) \neq \mathbf{y}^s]$ plus the distribution discrepancy $\operatorname{disc}(P, Q)$ quantified by a novel conditional domain discriminator.

Adversarial learning, the key idea to enabling Generative Adversarial Networks (GANs) [17], has been successfully explored to minimize the cross-domain discrepancy [13, 51]. Denote by $\mathbf{f} = F(\mathbf{x})$ the feature representation and by $\mathbf{g} = G(\mathbf{x})$ the classifier prediction generated from the deep network $G$. Domain adversarial neural network (DANN) [13] is a two-player game: the first player is the domain discriminator $D$ trained to distinguish the source domain from the target domain and the second player is the feature representation $F$ trained simultaneously to confuse the domain discriminator $D$. The error function of the domain discriminator corresponds well to the discrepancy between feature distributions $P(\mathbf{f})$ and $Q(\mathbf{f})$ [12], a key to bound the target risk in the domain adaptation theory [4].

## 3.1 Conditional Discriminator

We further improve existing adversarial domain adaptation methods [12, 52, 51] in two directions. First, when the joint distributions of feature and class, i.e. $P(\mathbf{x}^s, \mathbf{y}^s)$ and $Q(\mathbf{x}^t, \mathbf{y}^t)$, are non-identical across domains, adapting only the feature representation $\mathbf{f}$ may be insufficient. Due to a quantitative study [54], deep representations eventually transition from general to specific along deep networks, with transferability decreased remarkably in the domain-specific feature layer $\mathbf{f}$ and classifier layer $\mathbf{g}$. Second, when the feature distribution is multimodal, which is a real scenario due to the nature of multi-class classification, adapting only the feature representation may be challenging for adversarial networks. Recent work [17, 2, 7, 1] reveals the high risk of failure in matching only a fraction of components underlying different distributions with the adversarial networks. Even if the discriminator is fully confused, we have no theoretical guarantee that two different distributions are identical [3].

This paper tackles the two aforementioned challenges by formalizing a conditional adversarial domain adaptation framework. Recent advances in Conditional Generative Adversarial Networks (CGANs) [34] discover that different distributions can be matched better by conditioning the generator and discriminator on relevant information, such as associated labels and affiliated modality. Conditional GANs [25, 35] generate globally coherent images from datasets with high variability and multimodal distributions. Motivated by conditional GANs, we observe that in adversarial domain adaptation, the classifier prediction $\mathbf{g}$ conveys discriminative information potentially revealing the multimodal structures, which can be conditioned on when adapting feature representation $\mathbf{f}$. By conditioning, domain variances in both feature representation $\mathbf{f}$ and classifier prediction $\mathbf{g}$ can be modeled simultaneously.

We formulate Conditional Domain Adversarial Network (CDAN) as a minimax optimization problem with two competitive error terms: (a) $\mathcal{E}(G)$ on the source classifier $G$, which is minimized to guarantee lower source risk; (b) $\mathcal{E}(D, G)$ on the source classifier $G$ and the domain discriminator $D$ across the source and target domains, which is minimized over $D$ but maximized over $\mathbf{f} = F(\mathbf{x})$ and $\mathbf{g} = G(\mathbf{x})$:

$$\mathcal{E}(G) = \mathbb{E}_{(\mathbf{x}_i^s, \mathbf{y}_i^s) \sim \mathcal{D}_s} L\left(G\left(\mathbf{x}_i^s\right), \mathbf{y}_i^s\right), \tag{1}$$

$$\mathcal{E}(D, G) = -\mathbb{E}_{\mathbf{x}_i^s \sim \mathcal{D}_s} \log\left[D\left(\mathbf{f}_i^s, \mathbf{g}_i^s\right)\right] - \mathbb{E}_{\mathbf{x}_j^t \sim \mathcal{D}_t} \log\left[1 - D\left(\mathbf{f}_j^t, \mathbf{g}_j^t\right)\right], \tag{2}$$

where $L(\cdot, \cdot)$ is the cross-entropy loss, and $\mathbf{h} = (\mathbf{f}, \mathbf{g})$ is the joint variable of feature representation $\mathbf{f}$ and classifier prediction $\mathbf{g}$. The minimax game of conditional domain adversarial network (CDAN) is

$$
\begin{aligned}
&\min_{G} \ \mathcal{E}(G) - \lambda \mathcal{E}(D, G) \\
&\min_{D} \ \mathcal{E}(D, G),
\end{aligned}
\tag{3}
$$

where $\lambda$ is a hyper-parameter between the two objectives to tradeoff source risk and domain adversary.

We condition domain discriminator $D$ on the classifier prediction $\mathbf{g}$ through joint variable $\mathbf{h} = (\mathbf{f}, \mathbf{g})$. This conditional domain discriminator can potentially tackle the two aforementioned challenges of

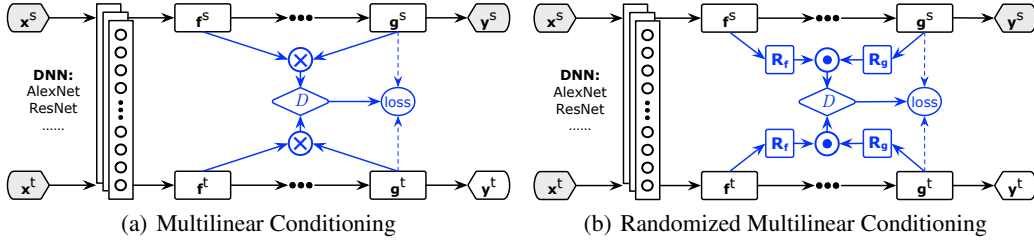

<div align="center">(a) Multilinear Conditioning          (b) Randomized Multilinear Conditioning</div>

Figure 1: Architectures of Conditional Domain Adversarial Networks (**CDAN**) for domain adaptation, where domain-specific feature representation $\mathbf{f}$ and classifier prediction $\mathbf{g}$ embody the cross-domain gap to be reduced jointly by the conditional domain discriminator $D$. (a) Multilinear (M) Conditioning, applicable to lower-dimensional scenario, where $D$ is conditioned on classifier prediction $\mathbf{g}$ via multilinear map $\mathbf{f} \otimes \mathbf{g}$; (b) Randomized Multilinear (RM) Conditioning, fit to higher-dimensional scenario, where $D$ is conditioned on classifier prediction $\mathbf{g}$ via randomized multilinear map $\frac{1}{\sqrt{d}}(\mathbf{R_f f}) \odot (\mathbf{R_g g})$. Entropy Conditioning (dashed line) leads to **CDAN+E** that prioritizes $D$ on easy-to-transfer examples.

adversarial domain adaptation. A simple conditioning of $D$ is $D(\mathbf{f} \oplus \mathbf{g})$, where we concatenate the feature representation and classifier prediction in vector $\mathbf{f} \oplus \mathbf{g}$ and feed it to conditional domain discriminator $D$. This conditioning strategy is widely adopted by existing conditional GANs [34, 25, 35]. However, with the concatenation strategy, $\mathbf{f}$ and $\mathbf{g}$ are independent on each other, thus failing to fully capture multiplicative interactions between feature representation and classifier prediction, which are crucial to domain adaptation. As a result, the multimodal information conveyed in classifier prediction cannot be fully exploited to match the multimodal distributions of complex domains [47].

## 3.2 Multilinear Conditioning

The multilinear map is defined as the outer product of multiple random vectors. And the multilinear map of infinite-dimensional nonlinear feature maps has been successfully applied to embed joint distribution or conditional distribution into reproducing kernel Hilbert spaces [47, 44, 45, 30]. Given two random vectors $\mathbf{x}$ and $\mathbf{y}$, the joint distribution $P(\mathbf{x}, \mathbf{y})$ can be modeled by the *cross-covariance* $\mathbb{E}_{\mathbf{xy}}[\phi(\mathbf{x}) \otimes \phi(\mathbf{y})]$, where $\phi$ is a feature map induced by some reproducing kernel. Such kernel embeddings enable manipulation of the multiplicative interactions across multiple random variables.

Besides the theoretical benefit of the multilinear map $\mathbf{x} \otimes \mathbf{y}$ over the concatenation $\mathbf{x} \oplus \mathbf{y}$ [47, 46], we further explain its superiority intuitively. Assume linear map $\phi(\mathbf{x}) = \mathbf{x}$ and one-hot label vector $\mathbf{y}$ in $C$ classes. As can be verified, mean map $\mathbb{E}_{\mathbf{xy}}[\mathbf{x} \oplus \mathbf{y}] = \mathbb{E}_{\mathbf{x}}[\mathbf{x}] \oplus \mathbb{E}_{\mathbf{y}}[\mathbf{y}]$ computes the means of $\mathbf{x}$ and $\mathbf{y}$ independently. In contrast, mean map $\mathbb{E}_{\mathbf{xy}}[\mathbf{x} \otimes \mathbf{y}] = \mathbb{E}_{\mathbf{x}}[\mathbf{x}|y = 1] \oplus \ldots \oplus \mathbb{E}_{\mathbf{x}}[\mathbf{x}|y = C]$ computes the means of each of the $C$ class-conditional distributions $P(\mathbf{x}|y)$. Superior than concatenation, the multilinear map $\mathbf{x} \otimes \mathbf{y}$ can fully capture the multimodal structures behind complex data distributions.

Taking the advantage of multilinear map, in this paper, we condition $D$ on $\mathbf{g}$ with the multilinear map

$$T_{\otimes} (\mathbf{f}, \mathbf{g}) = \mathbf{f} \otimes \mathbf{g}, \tag{4}$$

where $T_{\otimes}$ is a multilinear map and $D(\mathbf{f}, \mathbf{g}) = D(\mathbf{f} \otimes \mathbf{g})$. As such, the conditional domain discriminator successfully models the multimodal information and joint distributions of $\mathbf{f}$ and $\mathbf{g}$. Also, the multi-linearity can accommodate random vectors $\mathbf{f}$ and $\mathbf{g}$ with different cardinalities and magnitudes.

A disadvantage of the multilinear map is dimension explosion. Denoting by $d_f$ and $d_g$ the dimensions of vectors $\mathbf{f}$ and $\mathbf{g}$ respectively, the dimension of multilinear map $\mathbf{f} \otimes \mathbf{g}$ is $d_f \times d_g$, often too high-dimensional to be embedded into deep networks without causing parameter explosion. This paper addresses the dimension explosion by randomized methods [40, 26]. Note that multilinear map holds

$$\begin{aligned} \langle T_{\otimes} (\mathbf{f}, \mathbf{g}), T_{\otimes} (\mathbf{f}', \mathbf{g}') \rangle &= \langle \mathbf{f} \otimes \mathbf{g}, \mathbf{f}' \otimes \mathbf{g}' \rangle \\ &= \langle \mathbf{f}, \mathbf{f}' \rangle \langle \mathbf{g}, \mathbf{g}' \rangle \\ &\approx \langle T_{\odot} (\mathbf{f}, \mathbf{g}), T_{\odot} (\mathbf{f}', \mathbf{g}') \rangle, \end{aligned} \tag{5}$$

where $T_{\odot} (\mathbf{f}, \mathbf{g})$ is the explicit randomized multilinear map of dimension $d \ll d_f \times d_g$. We define

$$T_{\odot} (\mathbf{f}, \mathbf{g}) = \frac{1}{\sqrt{d}} (\mathbf{R_f f}) \odot (\mathbf{R_g g}), \tag{6}$$

where $\odot$ is element-wise product, $\mathbf{R_f}$ and $\mathbf{R_g}$ are random matrices sampled only once and fixed in training, and each element $R_{ij}$ follows a symmetric distribution with univariance, i.e. $\mathbb{E}[R_{ij}] = 0, \mathbb{E}[R_{ij}^2] = 1$. Applicable distributions include Gaussian distribution and Uniform distribution. As the inner-product on $T_\otimes$ can be accurately approximated by the inner-product on $T_\odot$, we can directly adopt $T_\odot(\mathbf{f}, \mathbf{g})$ for computation efficiency. We guarantee such approximation quality by a theorem.

**Theorem 1.** *The expectation and variance of using $T_\odot(\mathbf{f}, \mathbf{g})$ (6) to approximate $T_\otimes(\mathbf{f}, \mathbf{g})$ (4) satisfy*

$$
\mathbb{E}\left[\langle T_\odot(\mathbf{f}, \mathbf{g}), T_\odot(\mathbf{f}', \mathbf{g}')\rangle\right] = \langle \mathbf{f}, \mathbf{f}'\rangle \langle \mathbf{g}, \mathbf{g}'\rangle,
$$
$$
\mathrm{var}\left[\langle T_\odot(\mathbf{f}, \mathbf{g}), T_\odot(\mathbf{f}', \mathbf{g}')\rangle\right] = \sum_{i=1}^d \beta(\mathbf{R}_i^\mathbf{f}, \mathbf{f}) \beta(\mathbf{R}_i^\mathbf{g}, \mathbf{g}) + C, \tag{7}
$$

*where $\beta(\mathbf{R}_i^\mathbf{f}, \mathbf{f}) = \frac{1}{d}\sum_{j=1}^{d_f}[f_j^2 f_j'^2 \mathbb{E}[(R_{ij}^f)^4] + C']$ and similarly for $\beta(\mathbf{R}_i^\mathbf{g}, \mathbf{g})$, $C$, $C'$ are constants.*

*Proof.* The proof is given in the supplemental material. □

This verifies that $T_\odot$ is an unbiased estimate of $T_\otimes$ in terms of inner product, with estimation variance depending only on the fourth-order moments $\mathbb{E}[(R_{ij}^f)^4]$ and $\mathbb{E}[(R_{ij}^g)^4]$, which are constants for many symmetric distributions with univariance, including Gaussian distribution and Uniform distribution. The bound reveals that wen can further minimize the approximation error by normalizing the features.

For simplicity we define the conditioning strategy used by the conditional domain discriminator $D$ as

$$
T(\mathbf{h}) = \begin{cases} T_\otimes(\mathbf{f}, \mathbf{g}) & \text{if } d_f \times d_g \leqslant 4096 \\ T_\odot(\mathbf{f}, \mathbf{g}) & \text{otherwise,} \end{cases} \tag{8}
$$

where 4096 is the largest number of units in typical deep networks (e.g. AlexNet), and if dimension of the multilinear map $T_\otimes$ is larger than 4096, then we will choose randomized multilinear map $T_\odot$.

## 3.3 Conditional Domain Adversarial Network

We enable conditional adversarial domain adaptation over domain-specific feature representation $\mathbf{f}$ and classifier prediction $\mathbf{g}$. We jointly minimize (1) w.r.t. source classifier $G$ and feature extractor $F$, minimize (2) w.r.t. domain discriminator $D$, and maximize (2) w.r.t. feature extractor $F$ and source classifier $G$. This yields the minimax problem of Conditional Domain Adversarial Network (**CDAN**):

$$
\min_G \mathbb{E}_{(\mathbf{x}_i^s, \mathbf{y}_i^s) \sim \mathcal{D}_s} L(G(\mathbf{x}_i^s), \mathbf{y}_i^s)
$$
$$
+ \lambda \left( \mathbb{E}_{\mathbf{x}_i^s \sim \mathcal{D}_s} \log[D(T(\mathbf{h}_i^s))] + \mathbb{E}_{\mathbf{x}_j^t \sim \mathcal{D}_t} \log[1 - D(T(\mathbf{h}_j^t))] \right) \tag{9}
$$
$$
\max_D \mathbb{E}_{\mathbf{x}_i^s \sim \mathcal{D}_s} \log[D(T(\mathbf{h}_i^s))] + \mathbb{E}_{\mathbf{x}_j^t \sim \mathcal{D}_t} \log[1 - D(T(\mathbf{h}_j^t))],
$$

where $\lambda$ is a hyper-parameter between source classifier and conditional domain discriminator, and note that $\mathbf{h} = (\mathbf{f}, \mathbf{g})$ is the joint variable of domain-specific feature representation $\mathbf{f}$ and classifier prediction $\mathbf{g}$ for adversarial adaptation. As a rule of thumb, we can safely set $\mathbf{f}$ as the last feature layer representation and $\mathbf{g}$ as the classifier layer prediction. In cases where lower-layer features are not transferable as in pixel-level adaptation tasks [25, 22], we can change $\mathbf{f}$ to lower-layer representations.

**Entropy Conditioning** The minimax problem for the conditional domain discriminator (9) imposes equal importance for different examples, while hard-to-transfer examples with uncertain predictions may deteriorate the conditional adversarial adaptation procedure. Towards safe transfer, we quantify the uncertainty of classifier predictions by the entropy criterion $H(\mathbf{g}) = -\sum_{c=1}^C g_c \log g_c$, where $C$ is the number of classes and $g_c$ is the probability of predicting an example to class $c$. We prioritize the discriminator on those easy-to-transfer examples with certain predictions by reweighting each training example of the conditional domain discriminator by an entropy-aware weight $w(H(\mathbf{g})) = 1 + e^{-H(\mathbf{g})}$. The entropy conditioning variant of CDAN (**CDAN+E**) for improved transferability is formulated as

$$
\min_G \mathbb{E}_{(\mathbf{x}_i^s, \mathbf{y}_i^s) \sim \mathcal{D}_s} L(G(\mathbf{x}_i^s), \mathbf{y}_i^s)
$$
$$
+ \lambda \left( \mathbb{E}_{\mathbf{x}_i^s \sim \mathcal{D}_s} w(H(\mathbf{g}_i^s)) \log[D(T(\mathbf{h}_i^s))] + \mathbb{E}_{\mathbf{x}_j^t \sim \mathcal{D}_t} w(H(\mathbf{g}_j^t)) \log[1 - D(T(\mathbf{h}_j^t))] \right) \tag{10}
$$
$$
\max_D \mathbb{E}_{\mathbf{x}_i^s \sim \mathcal{D}_s} w(H(\mathbf{g}_i^s)) \log[D(T(\mathbf{h}_i^s))] + \mathbb{E}_{\mathbf{x}_j^t \sim \mathcal{D}_t} w(H(\mathbf{g}_j^t)) \log[1 - D(T(\mathbf{h}_j^t))].
$$

The domain discriminator empowers the entropy minimization principle [19] and encourages certain predictions, enabling CDAN+E to further perform semi-supervised learning on unlabeled target data.

### 3.4 Generalization Error Analysis

We give an analysis of the CDAN method taking similar formalism of the domain adaptation theory [5, 4]. We first consider the source and target domains over the fixed representation space $\mathbf{f} = F(\mathbf{x})$, and a family of source classifiers $G$ in hypothesis space $\mathcal{H}$ [13]. Denote by $\epsilon_P(G) = \mathbb{E}_{(\mathbf{f},\mathbf{y})\sim P}[G(\mathbf{f}) \neq \mathbf{y}]$ the risk of a hypothesis $G \in \mathcal{H}$ w.r.t. distribution $P$, and $\epsilon_P(G_1, G_2) = \mathbb{E}_{(\mathbf{f},\mathbf{y})\sim P}[G_1(\mathbf{f}) \neq G_2(\mathbf{f})]$ the disagreement between hypotheses $G_1, G_2 \in \mathcal{H}$. Let $G^* = \arg\min_G \epsilon_P(G) + \epsilon_Q(G)$ be the ideal hypothesis that explicitly embodies the notion of adaptability. The probabilistic bound [4] of the target risk $\epsilon_Q(G)$ of hypothesis $G$ is given by the source risk $\epsilon_P(G)$ plus the distribution discrepancy

$$\epsilon_Q(G) \leqslant \epsilon_P(G) + [\epsilon_P(G^*) + \epsilon_Q(G^*)] + |\epsilon_P(G, G^*) - \epsilon_Q(G, G^*)|. \tag{11}$$

The goal of domain adaptation is to reduce the distribution discrepancy $|\epsilon_P(G, G^*) - \epsilon_Q(G, G^*)|$.

By definition, $\epsilon_P(G, G^*) = \mathbb{E}_{(\mathbf{f},\mathbf{y})\sim P}[G(\mathbf{f}) \neq G^*(\mathbf{f})] = \mathbb{E}_{(\mathbf{f},\mathbf{g})\sim P_G}[\mathbf{g} \neq G^*(\mathbf{f})] = \epsilon_{P_G}(G^*)$, and similarly, $\epsilon_Q(G, G^*) = \epsilon_{Q_G}(G^*)$. Note that, $P_G = (\mathbf{f}, G(\mathbf{f}))_{\mathbf{f}\sim P(\mathbf{f})}$ and $Q_G = (\mathbf{f}, G(\mathbf{f}))_{\mathbf{f}\sim Q(\mathbf{f})}$ are the proxies of the joint distributions $P(\mathbf{f}, \mathbf{y})$ and $Q(\mathbf{f}, \mathbf{y})$, respectively [10]. Based on the proxies, $|\epsilon_P(G, G^*) - \epsilon_Q(G, G^*)| = |\epsilon_{P_G}(G^*) - \epsilon_{Q_G}(G^*)|$. Define a (loss) difference hypothesis space $\Delta \triangleq \{\delta = |\mathbf{g} - G^*(\mathbf{f})| : G^* \in \mathcal{H}\}$ over the joint variable $(\mathbf{f}, \mathbf{g})$, where $\delta : (\mathbf{f}, \mathbf{g}) \mapsto \{0, 1\}$ outputs the loss of $G^* \in \mathcal{H}$. Based on the above difference hypothesis space $\Delta$, we define the $\Delta$-distance as

$$
\begin{aligned}
d_\Delta(P_G, Q_G) &\triangleq \sup_{\delta \in \Delta} \left| \mathbb{E}_{(\mathbf{f},\mathbf{g})\sim P_G}[\delta(\mathbf{f}, \mathbf{g}) \neq 0] - \mathbb{E}_{(\mathbf{f},\mathbf{g})\sim Q_G}[\delta(\mathbf{f}, \mathbf{g}) \neq 0] \right| \\
&= \sup_{G^* \in \mathcal{H}} \left| \mathbb{E}_{(\mathbf{f},\mathbf{g})\sim P_G}[|\mathbf{g} - G^*(\mathbf{f})| \neq 0] - \mathbb{E}_{(\mathbf{f},\mathbf{g})\sim Q_G}[|\mathbf{g} - G^*(\mathbf{f})| \neq 0] \right| \\
&\geqslant \left| \mathbb{E}_{(\mathbf{f},\mathbf{g})\sim P_G}[\mathbf{g} \neq G^*(\mathbf{f})] - \mathbb{E}_{(\mathbf{f},\mathbf{g})\sim Q_G}[\mathbf{g} \neq G^*(\mathbf{f})] \right| = |\epsilon_{P_G}(G^*) - \epsilon_{Q_G}(G^*)|.
\end{aligned}
\tag{12}
$$

Hence, the domain discrepancy $|\epsilon_P(G, G^*) - \epsilon_Q(G, G^*)|$ can be upper-bounded by the $\Delta$-distance.

Since the difference hypothesis space $\Delta$ is a continuous function class, assume the family of domain discriminators $\mathcal{H}_D$ is rich enough to contain $\Delta$, $\Delta \subset \mathcal{H}_D$. Such an assumption is not unrealistic as we have the freedom to choose $\mathcal{H}_D$, for example, a multilayer perceptrons that can fit any functions. Given these assumptions, we show that training domain discriminator $D$ is related to $d_\Delta(P_G, Q_G)$:

$$
\begin{aligned}
d_\Delta(P_G, Q_G) &\leqslant \sup_{D \in \mathcal{H}_D} \left| \mathbb{E}_{(\mathbf{f},\mathbf{g})\sim P_G}[D(\mathbf{f}, \mathbf{g}) \neq 0] - \mathbb{E}_{(\mathbf{f},\mathbf{g})\sim Q_G}[D(\mathbf{f}, \mathbf{g}) \neq 0] \right| \\
&\leqslant \sup_{D \in \mathcal{H}_D} \left| \mathbb{E}_{(\mathbf{f},\mathbf{g})\sim P_G}[D(\mathbf{f}, \mathbf{g}) = 1] + \mathbb{E}_{(\mathbf{f},\mathbf{g})\sim Q_G}[D(\mathbf{f}, \mathbf{g}) = 0] \right|.
\end{aligned}
\tag{13}
$$

This supremum is achieved in the process of training the optimal discriminator $D$ in CDAN, giving an upper bound of $d_\Delta(P_G, Q_G)$. Simultaneously, we learn representation $\mathbf{f}$ to minimize $d_\Delta(P_G, Q_G)$, yielding better approximation of $\epsilon_Q(G)$ by $\epsilon_P(G)$ to bound the target risk in the minimax paradigm.

## 4 Experiments

We evaluate the proposed conditional domain adversarial networks with many state-of-the-art transfer learning and deep learning methods. Codes will be available at `http://github.com/thuml/CDAN`.

### 4.1 Setup

**Office-31** [42] is the most widely used dataset for visual domain adaptation, with 4,652 images and 31 categories collected from three distinct domains: *Amazon* (**A**), *Webcam* (**W**) and *DSLR* (**D**). We evaluate all methods on six transfer tasks $\mathbf{A} \to \mathbf{W}$, $\mathbf{D} \to \mathbf{W}$, $\mathbf{W} \to \mathbf{D}$, $\mathbf{A} \to \mathbf{D}$, $\mathbf{D} \to \mathbf{A}$, and $\mathbf{W} \to \mathbf{A}$.

**ImageCLEF-DA**[1] is a dataset organized by selecting the 12 common classes shared by three public datasets (domains): *Caltech-256* (**C**), *ImageNet ILSVRC 2012* (**I**), and *Pascal VOC 2012* (**P**). We permute all three domains and build six transfer tasks: $\mathbf{I} \to \mathbf{P}$, $\mathbf{P} \to \mathbf{I}$, $\mathbf{I} \to \mathbf{C}$, $\mathbf{C} \to \mathbf{I}$, $\mathbf{C} \to \mathbf{P}$, $\mathbf{P} \to \mathbf{C}$.

**Office-Home** [53] is a better organized but more difficult dataset than *Office-31*, which consists of 15,500 images in 65 object classes in office and home settings, forming four extremely dissimilar domains: Artistic images (**Ar**), Clip Art (**Cl**), Product images (**Pr**), and Real-World images (**Rw**).

**Digits** We investigate three digits datasets: **MNIST**, **USPS**, and Street View House Numbers (**SVHN**). We adopt the evaluation protocol of CyCADA [22] with three transfer tasks: USPS to MNIST (**U** $\to$ **M**), MNIST to USPS (**M** $\to$ **U**), and SVHN to MNIST (**S** $\to$ **M**). We train our model using the training sets: MNIST (60,000), USPS (7,291), standard SVHN train (73,257). Evaluation is reported on the standard test sets: MNIST (10,000), USPS (2,007) (the numbers of images are in parentheses).

**VisDA-2017**[2] is a challenging simulation-to-real dataset, with two very distinct domains: **Synthetic**, renderings of 3D models from different angles and with different lightning conditions; **Real**, natural images. It contains over 280K images across 12 classes in the training, validation and test domains.

We compare Conditional Domain Adversarial Network (**CDAN**) with state-of-art domain adaptation methods: Deep Adaptation Network (**DAN**) [29], Residual Transfer Network (**RTN**) [31], Domain Adversarial Neural Network (**DANN**) [13], Adversarial Discriminative Domain Adaptation (**ADDA**) [51], Joint Adaptation Network (**JAN**) [30], Unsupervised Image-to-Image Translation (**UNIT**) [28], Generate to Adapt (**GTA**) [43], Cycle-Consistent Adversarial Domain Adaptation (**CyCADA**) [22].

We follow the standard protocols for unsupervised domain adaptation [12, 30]. We use all labeled source examples and all unlabeled target examples and compare the average classification accuracy based on three random experiments. We conduct importance-weighted cross-validation (**IWCV**) [48] to select hyper-parameters for all methods. As CDAN performs stably under different parameters, we fix $\lambda = 1$ for all experiments. For MMD-based methods (TCA, DAN, RTN, and JAN), we use Gaussian kernel with bandwidth set to median pairwise distances on training data [29]. We adopt **AlexNet** [27] and **ResNet-50** [20] as base networks and all methods differ only in their discriminators.

We implement AlexNet-based methods in **Caffe** and ResNet-based methods in **PyTorch**. We fine-tune from ImageNet pre-trained models [41], except the digit datasets that we train our models from scratch. We train the new layers and classifier layer through back-propagation, where the classifier is trained from scratch with learning rate 10 times that of the lower layers. We adopt mini-batch SGD with momentum of 0.9 and the learning rate annealing strategy as [13]: the learning rate is adjusted by $\eta_p = \eta_0(1 + \alpha p)^{-\beta}$, where $p$ is the training progress changing from 0 to 1, and $\eta_0 = 0.01, \alpha = 10, \beta = 0.75$ are optimized by the importance-weighted cross-validation [48]. We adopt a progressive training strategy for the discriminator, increasing $\lambda$ from 0 to 1 by multiplying to $\frac{1-\exp(-\delta p)}{1+\exp(-\delta p)}, \delta = 10$.

Table 1: Accuracy (%) on Office-31 for unsupervised domain adaptation (AlexNet and ResNet)

| Method | A → W | D → W | W → D | A → D | D → A | W → A | Avg |
|---|---|---|---|---|---|---|---|
| AlexNet [27] | 61.6±0.5 | 95.4±0.3 | 99.0±0.2 | 63.8±0.5 | 51.1±0.6 | 49.8±0.4 | 70.1 |
| DAN [29] | 68.5±0.5 | 96.0±0.3 | 99.0±0.3 | 67.0±0.4 | 54.0±0.5 | 53.1±0.5 | 72.9 |
| RTN [31] | 73.3±0.3 | 96.8±0.2 | 99.6±0.1 | 71.0±0.2 | 50.5±0.3 | 51.0±0.1 | 73.7 |
| DANN [13] | 73.0±0.5 | 96.4±0.3 | 99.2±0.3 | 72.3±0.3 | 53.4±0.4 | 51.2±0.5 | 74.3 |
| ADDA [51] | 73.5±0.6 | 96.2±0.4 | 98.8±0.4 | 71.6±0.4 | 54.6±0.5 | 53.5±0.6 | 74.7 |
| JAN [30] | 74.9±0.3 | 96.6±0.2 | 99.5±0.2 | 71.8±0.2 | **58.3**±0.3 | 55.0±0.4 | 76.0 |
| **CDAN** | 77.9±0.3 | 96.9±0.2 | **100.0**±.0 | 75.1±0.2 | 54.5±0.3 | **57.5**±0.4 | 77.0 |
| **CDAN+E** | **78.3**±0.2 | **97.2**±0.1 | **100.0**±.0 | **76.3**±0.1 | 57.3±0.2 | 57.3±0.3 | **77.7** |
| ResNet-50 [20] | 68.4±0.2 | 96.7±0.1 | 99.3±0.1 | 68.9±0.2 | 62.5±0.3 | 60.7±0.3 | 76.1 |
| DAN [29] | 80.5±0.4 | 97.1±0.2 | 99.6±0.1 | 78.6±0.2 | 63.6±0.3 | 62.8±0.2 | 80.4 |
| RTN [31] | 84.5±0.2 | 96.8±0.1 | 99.4±0.1 | 77.5±0.3 | 66.2±0.2 | 64.8±0.3 | 81.6 |
| DANN [13] | 82.0±0.4 | 96.9±0.2 | 99.1±0.1 | 79.7±0.4 | 68.2±0.4 | 67.4±0.5 | 82.2 |
| ADDA [51] | 86.2±0.5 | 96.2±0.3 | 98.4±0.3 | 77.8±0.3 | 69.5±0.4 | 68.9±0.5 | 82.9 |
| JAN [30] | 85.4±0.3 | 97.4±0.2 | 99.8±0.2 | 84.7±0.3 | 68.6±0.3 | 70.0±0.4 | 84.3 |
| GTA [43] | 89.5±0.5 | 97.9±0.3 | 99.8±0.4 | 87.7±0.5 | **72.8**±0.3 | **71.4**±0.4 | 86.5 |
| **CDAN** | 93.1±0.2 | 98.2±0.2 | **100.0**±.0 | 89.8±0.3 | 70.1±0.4 | 68.0±0.4 | 86.6 |
| **CDAN+E** | **94.1**±0.1 | **98.6**±0.1 | **100.0**±.0 | **92.9**±0.2 | 71.0±0.3 | 69.3±0.3 | **87.7** |

## 4.2 Results

The results on *Office-31* based on AlexNet and ResNet are reported in Table 1, with results of baselines directly reported from the original papers if protocol is the same. The CDAN models significantly outperform all comparison methods on most transfer tasks, where CDAN+E is a top-performing variant and CDAN performs slightly worse. It is desirable that CDAN promotes the classification accuracies substantially on hard transfer tasks, e.g. **A → W** and **A → D**, where the source and target domains are substantially different [42]. Note that, CDAN+E even outperforms generative pixel-level domain adaptation method GTA, which has a very complex design in both architecture and objectives.

The results on the *ImageCLEF-DA* dataset are reported in Table 2. The CDAN models outperform the comparison methods on most transfer tasks, but with smaller rooms of improvement. This is reasonable since the three domains in *ImageCLEF-DA* are of equal size and balanced in each category, and are visually more similar than *Office-31*, making the former dataset easier for domain adaptation.

Table 2: Accuracy (%) on ImageCLEF-DA for unsupervised domain adaptation (AlexNet and ResNet)

| Method | I → P | P → I | I → C | C → I | C → P | P → C | Avg |
|---|---|---|---|---|---|---|---|
| AlexNet [27] | 66.2±0.2 | 70.0±0.2 | 84.3±0.2 | 71.3±0.4 | 59.3±0.5 | 84.5±0.3 | 73.9 |
| DAN [29] | 67.3±0.2 | 80.5±0.3 | 87.7±0.3 | 76.0±0.3 | 61.6±0.3 | 88.4±0.2 | 76.9 |
| DANN [13] | 66.5±0.6 | 81.8±0.3 | 89.0±0.4 | 79.8±0.6 | 63.5±0.5 | 88.7±0.3 | 78.2 |
| JAN [30] | 67.2±0.5 | 82.8±0.4 | 91.3±0.5 | 80.0±0.5 | 63.5±0.4 | 91.0±0.4 | 79.3 |
| **CDAN** | **67.7**±0.3 | 83.3±0.1 | 91.8±0.2 | **81.5**±0.2 | 63.0±0.2 | 91.5±0.3 | 79.8 |
| **CDAN+E** | 67.0±0.4 | **84.8**±0.2 | **92.4**±0.3 | 81.3±0.3 | **64.7**±0.3 | **91.6**±0.4 | **80.3** |
| ResNet-50 [20] | 74.8±0.3 | 83.9±0.1 | 91.5±0.3 | 78.0±0.2 | 65.5±0.3 | 91.2±0.3 | 80.7 |
| DAN [29] | 74.5±0.4 | 82.2±0.2 | 92.8±0.2 | 86.3±0.4 | 69.2±0.4 | 89.8±0.4 | 82.5 |
| DANN [13] | 75.0±0.6 | 86.0±0.3 | 96.2±0.4 | 87.0±0.5 | 74.3±0.5 | 91.5±0.6 | 85.0 |
| JAN [30] | 76.8±0.4 | 88.0±0.2 | 94.7±0.2 | 89.5±0.3 | 74.2±0.3 | 91.7±0.3 | 85.8 |
| **CDAN** | 76.7±0.3 | 90.6±0.3 | 97.0±0.4 | 90.5±0.4 | **74.5**±0.3 | 93.5±0.4 | 87.1 |
| **CDAN+E** | **77.7**±0.3 | **90.7**±0.2 | **97.7**±0.3 | **91.3**±0.3 | 74.2±0.2 | **94.3**±0.3 | **87.7** |

Table 3: Accuracy (%) on Office-Home for unsupervised domain adaptation (AlexNet and ResNet)

| Method | Ar→Cl | Ar→Pr | Ar→Rw | Cl→Ar | Cl→Pr | Cl→Rw | Pr→Ar | Pr→Cl | Pr→Rw | Rw→Ar | Rw→Cl | Rw→Pr | Avg |
|---|---|---|---|---|---|---|---|---|---|---|---|---|---|
| AlexNet [27] | 26.4 | 32.6 | 41.3 | 22.1 | 41.7 | 42.1 | 20.5 | 20.3 | 51.1 | 31.0 | 27.9 | 54.9 | 34.3 |
| DAN [29] | 31.7 | 43.2 | 55.1 | 33.8 | 48.6 | 50.8 | 30.1 | 35.1 | 57.7 | 44.6 | 39.3 | 63.7 | 44.5 |
| DANN [13] | 36.4 | 45.2 | 54.7 | 35.2 | 51.8 | 55.1 | 31.6 | 39.7 | 59.3 | 45.7 | 46.4 | 65.9 | 47.3 |
| JAN [30] | 35.5 | 46.1 | 57.7 | 36.4 | 53.3 | 54.5 | 33.4 | 40.3 | 60.1 | 45.9 | 47.4 | 67.9 | 48.2 |
| **CDAN** | 36.2 | 47.3 | 58.6 | 37.3 | 54.4 | **58.3** | 33.2 | **43.9** | 62.1 | 48.2 | 48.1 | 70.7 | 49.9 |
| **CDAN+E** | **38.1** | **50.3** | **60.3** | **39.7** | **56.4** | 57.8 | **35.5** | 43.1 | **63.2** | **48.4** | **48.5** | **71.1** | **51.0** |
| ResNet-50 [20] | 34.9 | 50.0 | 58.0 | 37.4 | 41.9 | 46.2 | 38.5 | 31.2 | 60.4 | 53.9 | 41.2 | 59.9 | 46.1 |
| DAN [29] | 43.6 | 57.0 | 67.9 | 45.8 | 56.5 | 60.4 | 44.0 | 43.6 | 67.7 | 63.1 | 51.5 | 74.3 | 56.3 |
| DANN [13] | 45.6 | 59.3 | 70.1 | 47.0 | 58.5 | 60.9 | 46.1 | 43.7 | 68.5 | 63.2 | 51.8 | 76.8 | 57.6 |
| JAN [30] | 45.9 | 61.2 | 68.9 | 50.4 | 59.7 | 61.0 | 45.8 | 43.4 | 70.3 | 63.9 | 52.4 | 76.8 | 58.3 |
| **CDAN** | 49.0 | 69.3 | 74.5 | 54.4 | 66.0 | 68.4 | 55.6 | 48.3 | 75.9 | 68.4 | 55.4 | 80.5 | 63.8 |
| **CDAN+E** | **50.7** | **70.6** | **76.0** | **57.6** | **70.0** | **70.0** | **57.4** | **50.9** | **77.3** | **70.9** | **56.7** | **81.6** | **65.8** |

Table 4: Accuracy (%) on Digits and VisDA-2017 for unsupervised domain adaptation (ResNet-50)

| Method | M → U | U → M | S → M | Avg | Method | Synthetic → Real |
|---|---|---|---|---|---|---|
| UNIT [28] | **96.0** | 93.6 | **90.5** | 93.4 | JAN [30] | 61.6 |
| CyCADA [22] | 95.6 | 96.5 | 90.4 | 94.2 | GTA [43] | 69.5 |
| **CDAN** | 93.9 | 96.9 | 88.5 | 93.1 | **CDAN** | 66.8 |
| **CDAN+E** | 95.6 | **98.0** | 89.2 | **94.3** | **CDAN+E** | **70.0** |

The results on *Office-Home* are reported in Table 3. The CDAN models substantially outperform the comparison methods on most transfer tasks, and with larger rooms of improvement. An interpretation is that the four domains in *Office-Home* are with more categories, are visually more dissimilar with each other, and are difficult in each domain with much lower in-domain classification accuracy [53]. Since domain alignment is category agnostic in previous work, it is possible that the aligned domains are not classification friendly in the presence of large number of categories. It is desirable that CDAN models yield larger boosts on such difficult domain adaptation tasks, which highlights the power of adversarial domain adaptation by exploiting complex multimodal structures in classifier predictions.

Strong results are also achieved on the digits datasets and synthetic to real datasets as reported in Table 4. Note that the generative pixel-level adaptation methods UNIT, CyCADA, and GTA are specifically tailored to the digits and synthetic to real adaptation tasks. This explains why the previous feature-level adaptation method JAN performs fairly weakly. To our knowledge, CDAN+E is the only approach that works reasonably well on all five datasets, and remains a simple discriminative model.

### 4.3 Analysis

**Ablation Study** We examine the sampling strategies of the random matrices in Equation (6). We testify **CDAN+E (w/ gaussian sampling)** and **CDAN+E (w/ uniform sampling)** with their random matrices sampled only once from Gaussian and Uniform distributions, respectively. Table 5 shows that CDAN+E (w/o random sampling) performs best while CDAN+E (w/ uniform sampling) performs the best across the randomized variants. Table 1∼4 shows that **CDAN+E** outperforms **CDAN**, proving that entropy conditioning can prioritize easy-to-transfer examples and encourage certain predictions.

**Conditioning Strategies** Besides multilinear conditioning, we investigate **DANN-[f,g]** with the domain discriminator imposed on the concatenation of **f** and **g**, **DANN-f** and **DANN-g** with the

Table 5: Accuracy (%) of CDAN variants on Office-31 for unsupervised domain adaptation (ResNet)

| Method | A → W | D → W | W → D | A → D | D → A | W → A | Avg |
|---|---|---|---|---|---|---|---|
| CDAN+E (w/ gaussian sampling) | 93.0±0.2 | 98.4±0.2 | **100.0**±.0 | 89.2±0.3 | 70.2±0.4 | 67.4±0.4 | 86.4 |
| CDAN+E (w/ uniform sampling) | 94.0±0.2 | 98.4±0.2 | **100.0**±.0 | 89.8±0.3 | 70.1±0.4 | **69.4**±0.4 | 87.0 |
| CDAN+E (w/o random sampling) | **94.1**±0.1 | **98.6**±0.1 | **100.0**±.0 | **92.9**±0.2 | **71.0**±0.3 | 69.3±0.3 | **87.7** |

domain discriminator plugged in feature layer **f** and classifier layer **g**. Figure 2(a) shows accuracies on **A** → **W** and **A** → **D** based on ResNet-50. The concatenation strategy is not successful, as it cannot capture the cross-covariance between features and classes, which are crucial to domain adaptation [10]. Figure 2(b) shows that the entropy weight $e^{-H(\mathbf{g})}$ corresponds well with the prediction correctness: entropy weight $\approx 1$ if the prediction is correct, and much smaller than 1 when prediction is incorrect (uncertain). This reveals the power of the entropy conditioning to guarantee example transferability.

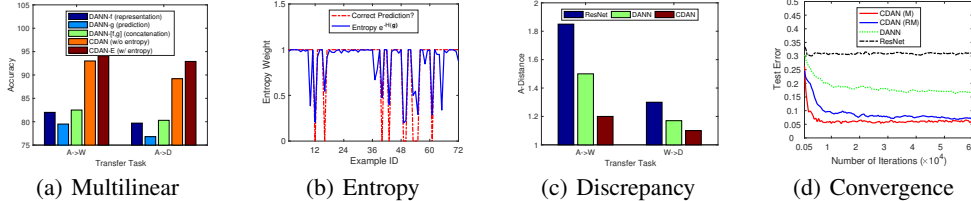

| (a) Multilinear | (b) Entropy | (c) Discrepancy | (d) Convergence |

Figure 2: Analysis of conditioning strategies, distribution discrepancy, and convergence.

**Distribution Discrepancy**    The $\mathcal{A}$-distance is a measure for distribution discrepancy [4, 33], defined as $\text{dist}_{\mathcal{A}} = 2\left(1 - 2\epsilon\right)$, where $\epsilon$ is the test error of a classifier trained to discriminate the source from target. Figure 2(c) shows $\text{dist}_{\mathcal{A}}$ on tasks **A** → **W**, **W** → **D** with features of ResNet, DANN, and CDAN. We observe that $\text{dist}_{\mathcal{A}}$ on CDAN features is smaller than $\text{dist}_{\mathcal{A}}$ on both ResNet and DANN features, implying that CDAN features can reduce the domain gap more effectively. As domains **W** and **D** are similar, $\text{dist}_{\mathcal{A}}$ of task **W** → **D** is smaller than that of **A** → **W**, implying higher accuracies.

**Convergence**    We testify the convergence of ResNet, DANN, and CDANs, with the test errors on task **A** → **W** shown in Figure 2(d). CDAN enjoys faster convergence than DANN, while CDAN (M) converges faster than CDAN (RM). Note that CDAN (M) constitutes high-dimensional multilinear map, which is slightly more costly than CDAN (RM), while CDAN (RM) has similar cost as DANN.

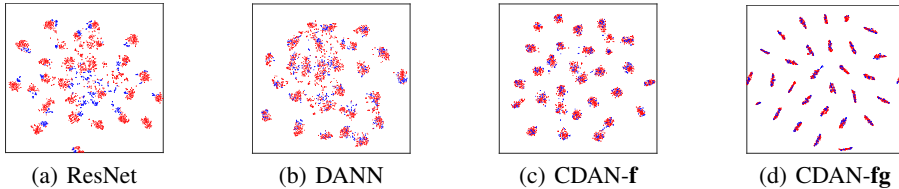

| (a) ResNet | (b) DANN | (c) CDAN-**f** | (d) CDAN-**fg** |

Figure 3: T-SNE of (a) ResNet, (b) DANN, (c) CDAN-**f**, (d) CDAN-**fg** (red: **A**; blue: **W**).

**Visualization**    We visualize by t-SNE [32] in Figures 3(a)–3(d) the representations of task **A** → **W** (31 classes) by ResNet, DANN, CDAN-f, and CDAN-fg. The source and target are not aligned well with ResNet, better aligned with DANN but categories are not discriminated well. They are aligned better and categories are discriminated better by CDAN-f, while CDAN-fg is evidently better than CDAN-f. This shows the benefit of conditioning adversarial adaptation on discriminative predictions.

# 5   Conclusion

This paper presented conditional domain adversarial networks (CDANs), novel approaches to domain adaptation with multimodal distributions. Unlike previous adversarial adaptation methods that solely match the feature representation across domains which is prone to under-matching, the proposed approach further conditions the adversarial domain adaptation on discriminative information to enable alignment of multimodal distributions. Experiments validated the efficacy of the proposed approach.

## Acknowledgments

We thank Yuchen Zhang at Tsinghua University for insightful discussions. This work was supported by the National Key R&D Program of China (2016YFB1000701), the Natural Science Foundation of China (61772299, 71690231, 61502265) and the DARPA Program on Lifelong Learning Machines.

## Footnotes

[1] `http://imageclef.org/2014/adaptation`

[2]http://ai.bu.edu/visda-2017/

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
