[Supplementary Material · CADA-supp.pdf]

# Supplemental Material: Conditional Adversarial Domain Adaptation

**Mingsheng Long[†], Zhangjie Cao[†], Jianmin Wang[†], and Michael I. Jordan[♯]**
[†]School of Software, Tsinghua University, China
[†]KLiss, MOE; BNRist; Research Center for Big Data, Tsinghua University, China
[♯]University of California, Berkeley, Berkeley, USA
{mingsheng,jimwang}@tsinghua.edu.cn caozhangjie14@gmail.com
jordan@berkeley.edu

## 1 Proof of Theorem 1

This supplemental material provides proof to Theorem 1 in the main paper. To enable better readability, denote by $\mathbf{z}^1 = \mathbf{f}$, $\mathbf{z}^2 = \mathbf{g}$ and $\mathbf{R}^1 = \mathbf{R_f}$ and $\mathbf{R}^2 = \mathbf{R_g}$. We first rewrite the randomized feature map

$$T_\odot(\mathbf{z}) = \frac{1}{\sqrt{d}} \left( \odot_\ell \mathbf{R}^\ell \mathbf{z}^\ell \right). \tag{1}$$

**Theorem 1.** *The expectation and variance of the inner products between the randomized feature maps $T_\odot(\mathbf{z})$ (1) generated by random matrices $\mathbf{R}^\ell, \ell = 1, 2$ satisfy*

$$\mathbb{E}\left[ \langle T_\odot(\mathbf{z}), T_\odot(\mathbf{z}') \rangle \right] = \prod_\ell \left\langle \mathbf{z}^\ell, \mathbf{z}'^\ell \right\rangle, \tag{2}$$

$$\mathrm{var}\left[ \langle T_\odot(\mathbf{z}), T_\odot(\mathbf{z}') \rangle \right] = \frac{1}{d^2} \sum_{i=1}^d \prod_\ell \left[ \sum_{j=1}^{d_\ell} \left( z_j^\ell \right)^2 \left( z_j'^\ell \right)^2 \mathbb{E}\left[ \left( R_{ij}^\ell \right)^4 \right] + C' \right] + C, \tag{3}$$

*where $C$ and $C'$ are constants that do not depend on the random matrices $\mathbf{R}^\ell, \ell = 1, 2$.*

Theorem 1 reveals that the inner product between the randomized feature maps $T_\odot(\mathbf{z})$ is an *unbiased* estimate of the inner product between the original multilinear fusions based on tensor products $T_\otimes(\mathbf{z})$. The variance of the inner product between the randomized feature maps $T_\odot(\mathbf{z})$ is depending only on the moments $\mathbb{E}[\left( R_{ij}^\ell \right)^4]$, which are constants for many symmetric distributions with univariance, i.e. $\mathbb{E}\left[ R_{ij}^\ell \right] = 0, \mathbb{E}[(R_{ij}^\ell)^2] = 1$. We can verify that: (1) for Bernoulli distribution, $\mathbb{E}\left( R_{ij}^\ell \right)^4 = 1$; (2) for standard normal distribution, $\mathbb{E}\left( R_{ij}^\ell \right)^4 = 3$; (3) for uniform distribution, $\mathbb{E}\left( R_{ij}^\ell \right)^4 = 1.8$. Therefore, for continuous sampling distributions, uniform distribution will yield the lowest estimation variance. The empirical study confirms that uniform distribution leads to the best multilinear fusion accuracy.

*Proof.*

$$
\begin{aligned}
\mathbb{E}\left[ \langle T_\odot(\mathbf{z}), T_\odot(\mathbf{z}') \rangle \right] &= \mathbb{E}\left[ \left\langle \frac{1}{\sqrt{d}} \left( \odot_\ell \mathbf{R}^\ell \mathbf{z}^\ell \right), \frac{1}{\sqrt{d}} \left( \odot_\ell \mathbf{R}^\ell \mathbf{z}'^\ell \right) \right\rangle \right] \\
&= \frac{1}{d} \mathbb{E}\left[ \left\langle \odot_\ell \mathbf{R}^\ell \mathbf{z}^\ell, \odot_\ell \mathbf{R}^\ell \mathbf{z}'^\ell \right\rangle \right] \\
&= \frac{1}{d} \mathbb{E}\left[ \sum_{i=1}^d \odot_\ell \left( \mathbf{R}_{i\cdot}^\ell \mathbf{z}^\ell \right) \left( \mathbf{R}_{i\cdot}^\ell \mathbf{z}'^\ell \right) \right] = \frac{1}{d} \sum_{i=1}^d \mathbb{E}\left[ \odot_\ell \left( \mathbf{R}_{i\cdot}^\ell \mathbf{z}^\ell \right) \left( \mathbf{R}_{i\cdot}^\ell \mathbf{z}'^\ell \right) \right] \\
&= \frac{1}{d} \sum_{i=1}^d \left[ \prod_\ell \mathbf{z}^{\ell\mathsf{T}} \mathbb{E}\left[ \mathbf{R}_{i\cdot}^{\ell\mathsf{T}} \mathbf{R}_{i\cdot}^\ell \right] \mathbf{z}'^\ell \right] = \frac{1}{d} \sum_{i=1}^d \left[ \prod_\ell \mathbf{z}^{\ell\mathsf{T}} \mathbf{z}'^\ell \right] = \prod_\ell \left\langle \mathbf{z}^\ell, \mathbf{z}'^\ell \right\rangle.
\end{aligned}
\tag{4}
$$

$$\text{var}\left[\langle T_\odot(\mathbf{z}), T_\odot(\mathbf{z}')\rangle\right] = \mathbb{E}\left[\langle T_\odot(\mathbf{z}), T_\odot(\mathbf{z}')\rangle^2\right] - \mathbb{E}[\langle T_\odot(\mathbf{z}), T_\odot(\mathbf{z}')\rangle]^2$$

$$= \mathbb{E}\left[\left\langle \frac{1}{\sqrt{d}}\left(\odot_\ell \mathbf{R}^\ell \mathbf{z}^\ell\right), \frac{1}{\sqrt{d}}\left(\odot_\ell \mathbf{R}^\ell \mathbf{z}'^\ell\right)\right\rangle^2\right] - \left(\prod_\ell \langle \mathbf{z}^\ell, \mathbf{z}'^\ell\rangle\right)^2$$

$$= \left[\frac{1}{d^2}\left(\sum_{i=1}^{d} \prod_\ell \left(\mathbf{R}_{i\cdot}^\ell \mathbf{z}^\ell\right)\left(\mathbf{R}_{i\cdot}^\ell \mathbf{z}'^\ell\right)\right)^2\right] - C_1$$

$$= \frac{1}{d^2}\mathbb{E}\left[\sum_{i=1}^{d}\left(\prod_\ell \left(\mathbf{R}_{i\cdot}^\ell \mathbf{z}^\ell\right)\left(\mathbf{R}_{i\cdot}^\ell \mathbf{z}'^\ell\right)\right)^2 + \sum_{i=1}^{d}\sum_{j\neq i}^{d}\left(\prod_\ell \left(\mathbf{R}_{i\cdot}^\ell \mathbf{z}^\ell\right)\left(\mathbf{R}_{i\cdot}^\ell \mathbf{z}'^\ell\right)\right)\left(\prod_\ell \left(\mathbf{R}_{j\cdot}^\ell \mathbf{z}^\ell\right)\left(\mathbf{R}_{j\cdot}^\ell \mathbf{z}'^\ell\right)\right)\right] - C_1$$

$$= \frac{1}{d^2}\sum_{i=1}^{d}\mathbb{E}\left[\left(\prod_\ell \left(\mathbf{R}_{i\cdot}^\ell \mathbf{z}^\ell\right)\left(\mathbf{R}_{i\cdot}^\ell \mathbf{z}'^\ell\right)\right)^2\right]$$
$$+ \frac{1}{d^2}\sum_{i=1}^{d}\sum_{j\neq i}^{d}\mathbb{E}\left[\left(\prod_\ell \left(\mathbf{R}_{i\cdot}^\ell \mathbf{z}^\ell\right)\left(\mathbf{R}_{i\cdot}^\ell \mathbf{z}'^\ell\right)\right)\left(\prod_\ell \left(\mathbf{R}_{j\cdot}^\ell \mathbf{z}^\ell\right)\left(\mathbf{R}_{j\cdot}^\ell \mathbf{z}'^\ell\right)\right)\right] - C_1$$

$$= \frac{1}{d^2}\sum_{i=1}^{d}\mathbb{E}\left[\left(\prod_\ell \left(\mathbf{R}_{i\cdot}^\ell \mathbf{z}^\ell\right)\left(\mathbf{R}_{i\cdot}^\ell \mathbf{z}'^\ell\right)\right)^2\right]$$
$$+ \frac{1}{d^2}\sum_{i=1}^{d}\sum_{j\neq i}^{d}\mathbb{E}\left[\left(\prod_\ell \left(\mathbf{R}_{i\cdot}^\ell \mathbf{z}^\ell\right)\left(\mathbf{R}_{i\cdot}^\ell \mathbf{z}'^\ell\right)\right)\right]\mathbb{E}\left[\left(\prod_\ell \left(\mathbf{R}_{j\cdot}^\ell \mathbf{z}^\ell\right)\left(\mathbf{R}_{j\cdot}^\ell \mathbf{z}'^\ell\right)\right)\right] - C_1$$

$$= \frac{1}{d^2}\sum_{i=1}^{d}\mathbb{E}\left[\left(\prod_\ell \left(\mathbf{R}_{i\cdot}^\ell \mathbf{z}^\ell\right)\left(\mathbf{R}_{i\cdot}^\ell \mathbf{z}'^\ell\right)\right)^2\right] + \frac{d(d-1)}{d^2}\left(\prod_\ell \langle \mathbf{z}^\ell, \mathbf{z}'^\ell\rangle\right)^2 - C_1$$

$$= \frac{1}{d^2}\sum_{i=1}^{d}\mathbb{E}\left[\left(\prod_\ell \mathbf{z}^{\ell\mathsf{T}}\left(\mathbf{R}_{i\cdot}^{\ell\mathsf{T}}\mathbf{R}_{i\cdot}^\ell\right)\mathbf{z}'^\ell\right)^2\right] + C_2 - C_1$$

$$= \frac{1}{d^2}\sum_{i=1}^{d}\mathbb{E}\left[\prod_\ell \mathbf{z}^{\ell\mathsf{T}}\left(\mathbf{R}_{i\cdot}^{\ell\mathsf{T}}\mathbf{R}_{i\cdot}^\ell\right)\mathbf{z}'^\ell \mathbf{z}'^{\ell\mathsf{T}}\left(\mathbf{R}_{i\cdot}^{\ell\mathsf{T}}\mathbf{R}_{i\cdot}^\ell\right)\mathbf{z}^\ell\right] + C_2 - C_1$$

$$= \frac{1}{d^2}\sum_{i=1}^{d}\prod_\ell \mathbf{z}^{\ell\mathsf{T}}\mathbb{E}\left[\left(\mathbf{R}_{i\cdot}^{\ell\mathsf{T}}\mathbf{R}_{i\cdot}^\ell\right)\mathbf{z}'^\ell \mathbf{z}'^{\ell\mathsf{T}}\left(\mathbf{R}_{i\cdot}^{\ell\mathsf{T}}\mathbf{R}_{i\cdot}^\ell\right)\right]\mathbf{z}^\ell + C_2 - C_1$$

$$= \frac{1}{d^2}\sum_{i=1}^{d}\prod_\ell \sum_{j=1}^{d_\ell}\sum_{k=1}^{d_\ell} z_j^{\ell\mathsf{T}}\mathbb{E}\left[\left(\mathbf{R}_{i\cdot}^{\ell\mathsf{T}}\mathbf{R}_{i\cdot}^\ell\right)\mathbf{z}'^\ell \mathbf{z}'^{\ell\mathsf{T}}\left(\mathbf{R}_{i\cdot}^{\ell\mathsf{T}}\mathbf{R}_{i\cdot}^\ell\right)\right]_{jk} z_k^\ell + C_2 - C_1$$

$$= \frac{1}{d^2}\sum_{i=1}^{d}\prod_\ell \left(\sum_{j=1}^{d_\ell} z_j^{\ell\mathsf{T}}\mathbb{E}\left[\left(\mathbf{R}_{i\cdot}^{\ell\mathsf{T}}\mathbf{R}_{i\cdot}^\ell\right)\mathbf{z}'^\ell \mathbf{z}'^{\ell\mathsf{T}}\left(\mathbf{R}_{i\cdot}^{\ell\mathsf{T}}\mathbf{R}_{i\cdot}^\ell\right)\right]_{jj} z_j^\ell\right.$$
$$\left. + \sum_{k\neq j}^{d_\ell} z_j^{\ell\mathsf{T}}\mathbb{E}\left[\left(\mathbf{R}_{i\cdot}^{\ell\mathsf{T}}\mathbf{R}_{i\cdot}^\ell\right)\mathbf{z}'^\ell \mathbf{z}'^{\ell\mathsf{T}}\left(\mathbf{R}_{i\cdot}^{\ell\mathsf{T}}\mathbf{R}_{i\cdot}^\ell\right)\right]_{jk} z_k^\ell\right) + C_2 - C_1$$

$$= \frac{1}{d^2}\sum_{i=1}^{d}\prod_\ell \left(\sum_{j=1}^{d_\ell} z_j^{\ell\mathsf{T}}\mathbb{E}\left[\left(\mathbf{R}_{i\cdot}^{\ell\mathsf{T}}\mathbf{R}_{i\cdot}^\ell\right)\mathbf{z}'^\ell \mathbf{z}'^{\ell\mathsf{T}}\left(\mathbf{R}_{i\cdot}^{\ell\mathsf{T}}\mathbf{R}_{i\cdot}^\ell\right)\right]_{jj} z_j^\ell + \sum_{j=1}^{d_\ell}\sum_{k\neq j}^{d_\ell} z_j^\ell z_k^\ell z_j'^\ell z_k'^\ell\right) + C_2 - C_1$$

$$= \frac{1}{d^2}\sum_{i=1}^{d}\prod_\ell \left(\sum_{j=1}^{d_\ell} \left(z_j^\ell\right)^2\left(z_j'^\ell\right)^2 \sum_{k=1}^{d_\ell}\mathbb{E}\left[\left(R_{ij}^\ell R_{ij}^\ell\right)\right]\left[\left(R_{ik}^\ell R_{ik}^\ell\right)\right] + C_3\right) + C_2 - C_1$$

$$= \frac{1}{d^2}\sum_{i=1}^{d}\prod_\ell \left(\sum_{j=1}^{d_\ell} \left(z_j^\ell\right)^2\left(z_j'^\ell\right)^2 \mathbb{E}\left[\left(R_{ij}^\ell\right)^4\right] + \sum_{j=1}^{d_\ell}\left(z_j^\ell\right)^2\left(z_j'^\ell\right)^2 \sum_{k\neq j}^{d_\ell} 1 + C_3\right) + C_2 - C_1$$

$$= \frac{1}{d^2}\sum_{i=1}^{d}\prod_\ell \left(\sum_{j=1}^{d_\ell} \left(z_j^\ell\right)^2\left(z_j'^\ell\right)^2 \mathbb{E}\left[\left(R_{ij}^\ell\right)^4\right] + C_4 + C_3\right) + C_2 - C_1$$

$$= \frac{1}{d^2}\sum_{i=1}^{d}\prod_\ell \left(\sum_{j=1}^{d_\ell} \left(z_j^\ell\right)^2\left(z_j'^\ell\right)^2 \mathbb{E}\left[\left(R_{ij}^\ell\right)^4\right] + C'\right) + C.$$

$$(5)$$

Since the equations in this proof look quite lengthy, we simplify the notations by denoting any parts of the equations independent on random matrices $\mathbf{R}^\ell, \ell = 1, 2$ as constants, such as $C_1 \sim C_4, C$, and $C'$. $\qquad\square$