[Reviews · NeurIPS 2018]

Reviewer 1



Quality and Clarity: The paper is an easy and pleasant read. It is well-organized and the claims are supported by either theoretical justifications or empirical results. The paper is technically sound. Originality: It proposes to bring in the idea of conditional GANs into adversarial domain adaptation. In doing so, it proposes a novel approach of explicit multilinear feature map to combine the feature representation as well as classifier output of a source/target sample before being fed to the domain-discriminator. Furthermore, it proposes a simple weighting for the samples in the adversarial loss. The weighting tries to up-weight “easy” samples in order to mitigate the vanishing gradient problem in the early stages of training. Significance: Although, the original contributions are quite simple and do not drastically change the formulation from the standard DANN [17], they are shown to be surprisingly and significantly effective in practice. The experiments are done on multiple hard benchmarks and further ablation analysis is conducted. The differences in performance and the accompanying analyses conclusively show the effectiveness of the proposed method compared to the prior works. Domain adaptation and adversarial training are important fields for both practitioners as well as academic researchers. The paper’s proposals are simple to adopt and clearly effective. Thus, it is a clear accept for NIPS. I do agree with R3: The rebuttal addresses most of reviewers' concerns. However, there is still an outstanding issue that I strongly encourage the authors to take into account for the next version: - The introduction should be toned down in claiming to be the first paper that considers conditional UDA. In that respect, the related works should include the discussions in the rebuttal on the difference with the works R3 mentions.

Reviewer 2



CDAN (Conditional Adversarial Domain Adaptation) is a new model for domain adaptation. Standard methods of DA use an adversarial setup where there is a network F that generate the features, a discriminator D that discriminates the features from Source and Target and a classifier G that ensures that the classification task is well done for the source domain. However the discriminator D only compares the features and ignores the output G. This article states that in the general classification task where there is multimode distributions the standard case can not perform well. Accordingly this paper uses a conditional GAN that takes into account into the discriminator both the features f and the output label g. The method propose several ways of combining f and g for conditioning the discriminator. Also the paper claims that there can be vanishing problems during the training and it shows that while the Wasserstein GAN loss solves it, it does not perform well for DA (I always wanted to know why Wasserstein loss was not used for DA). Accordingly it proposes a new loss that prevents the vanishing gradient and performs better: Priority loss. For combining f and g the paper proposes a method that can be intractable, accordingly they also propose an approximation. The method is tested in the standard DA tasks. However I miss results for the VISDA setup that lately is becoming a standard. The ablations studies are very meaningful: conditioning strategies, convergence, visualization and discrepancy. The paper is well written and motivated, the method explanation is clear and the formulation seams correct (However I didn't check it in depth and I skiped the demonstration of the theorem).

Reviewer 3



In this paper, the authors proposed a conditional adversarial domain adaptation framework, by considering a conditional discriminator which takes the multilinear maps of feature and network prediction as input. In addition, randomized multilinear conditioning was considered to reduce the input dimension with comparable performance. Overall, the presentation is clear and easy to follow. The proposed method is technically correct. However, I am leaning towards rejection since the contribution and novelty of this paper is limited. Having multimodal distributions in both source and target domains is a well-known challenge in domain adaptation. This paper is clearly not the first work to explicitly or implicitly address such challenge under a conditional adversarial framework. There are multiple previous related works, such as: Hoffman et al., FCNs in the Wild: Pixel-level Adversarial and Constraint-based Adaptation, arXiv16 Chen et al., No More Discrimination: Cross City Adaptation of Road Scene Segmenters, ICCV17 Tsai et al., Learning to Adapt Structured Output Space for Semantic Segmentation, CVPR18 In particular, the second work leverages network prediction for class-wise domain alignment, while the third work imposed discriminators on both features and network predictions. Both previous work share certain similarity to this work in terms of incorporating conditional information in the discriminator. However, none of the above works were cited or compared in this paper. The proposal of multilinear conditioning and randomized multilinear conditioning is a novel part of this work but the novelty/contribution doesn't seem enough for NIPS. In addition, this part lacks more principled theoretical analysis/ablation study why the proposed methods are better than other types of conditional input to discriminator. Experiment of this work lacks comparison to multiple state-of-the-arts and GAN baselines on unsupervised domain adaptation, including: Hoffman et al., CyCADA: Cycle-Consistent Adversarial Domain Adaptation, ICML18 Zhu et al., Unpaired Image-to-Image Translation using Cycle-Consistent Adversarial Networks, ICCV17 Liu et al., Unsupervised Image-to-Image Translation Networks, NIPS17 The authors could have conducted UDA experiments on several digit datasets following the above works. ================================================================================================ Updates: I have carefully looked into the rebuttal, the other reviews, and the draft paper (again). I understand the authors have put in good effort trying to convince me, which I appreciate very much. However, for now my overall view towards the paper remains similar to my initial decision. I want to be clear that I have no problem with the technical contents of the paper. To me the proposed method makes sense and seems technically correct. My major concern lies in that: If the paper is accepted to NIPS, it just makes another mediocre work that does not distinguish itself from the abundant closely related literature with clear insights. Here are my reasons: Within the context of this paper, one can decompose the contributions into two perspectives: 1. Discovering the fact that conditional information is used in adversarial training can help the UDA problem 2. Presenting a novel way how conditional information participates in adversarial training, and theoretically/empirically show why this is better than other proposed ways of incorporating conditional information participates. I understand the importance of using conditional information in adversarial training based UDA. In fact many previous work have similarly discovered the multi-modal distribution issue and accordingly presented different ways to incorporate conditional information. One big problem with this paper is that it is written in such as way that one who is not an expert in this field is likely to feel that this work has contribution in item 1. An example is the way how the introduction, and related work are written, where the authors claim they get the inspiration from conditional generative methods without mentioning mentioning other conditional UDA methods at all. The title of this paper also strengthened such impression. I understand the authors argued by emphasizing that their definition of "conditional adversarial" is actually based on item 2 not item 1 (the fact that both class label and feature are jointly taken as input to the same discriminator). However, in the original submission the authors mentioned that "Motivated by the conditioning insight, this paper presents Conditional Domain Adversarial Networks (CDAN) to exploit discriminative information conveyed in the classifier predictions to inform adversarial adaptation." There is no way one can be certain the definition is based on item 2. Such definition seems to fit perfectly to other conditional UDA methods as well. Finally, Assuming that the authors resolves the above issues by completely rewriting the introduction and related work, current form of this work is not enough to support item 2 even if the rebuttals are taken into consideration. The authors have shown HOW this method is different from other conditional UDA methods. However, the theoretical insights WHY this method is better over others seems below the NIPS standard. The only solid theoretical analysis falls on proving the good approximation of multilinear mapping by inner product, of which similar theories (unbias estimator) were also well-established by previous random feature methods. On the empirical side, the authors did not show any segmentation applications where considerable number of conditional UDA methods were proposed. The performance gain from the proposed multilinear conditional adversarial module (which is the most important claim of contribution in this paper) over other similar works seems marginal if the priority loss is not included. It is also unclear how the proposed method performs on the digits experiments without priority loss. In addition, details on the encoder architecture (ResNet-50? AlexNet? Others?) are missing too. It is not surprising these factors could all contribute to the performance gain, and currently the only information one has is that the authors achieved good system-level results with the help of priority loss. Besides the above concerns, for reference [1] in the rebuttal, the fact that the generation network jointly takes both classifier prediction and the feature as input to generate images that are further used as input to the discriminator also seems to fall into the author's definition of "conditional adversarial". The claim in the paper: "Towards adversarial learning for domain adaptation, only unconditional ones have been exploited while conditional ones remain unexplored." is therefore wrong. I think the authors showed some improvements on the tasks at system level. For this I decide to increase my overall score to 5. However, my feeling is that the authors' response can not convince me much on the above concerns.